# Near-Infrared Light-Triggered Nitric Oxide Nanogenerators for NO-Photothermal Synergistic Cancer Therapy

**DOI:** 10.3390/nano12081348

**Published:** 2022-04-14

**Authors:** Weiwei Liu, Farouk Semcheddine, Zengchao Guo, Hui Jiang, Xuemei Wang

**Affiliations:** 1State Key Laboratory of Bioelectronics, National Demonstration Center for Experimental Biomedical Engineering Education, School of Biological Science and Medical Engineering, Southeast University, Nanjing 210096, China; liuw182@163.com (W.L.); fou_semch@yahoo.fr (F.S.); rmcqh242526@163.com (Z.G.); 2School of Chemistry and Chemical Engineering, Southeast University, Nanjing 210096, China

**Keywords:** MOF, BNN6, gold nanoshells, NO-photothermal therapy

## Abstract

Cancer is still one of the major health issues faced by human beings today. Various nanomaterials have been designed to treat tumors and have made great progress. Herein, we used amino-functionalized metal organic framework (UiO-66-NH_2_) as superior templates and successfully synthesized the UiO-66-NH_2_@Au_shell_ composite nanoparticles (UA) with high loading capacity and excellent photothermal properties through a simple and gentle method. In addition, due to the rich pore structure and excellent biocompatibility of the as-prepared composite nanoparticles, the hydrophobic NO donor BNN6 (N,N′-Di-sec-butyl-N,N′-dinitroso-1, 4-phenylenediamine) molecule was efficiently delivered. Based on the phenomenon where BNN6 molecules can decompose and release NO at high temperature, when UiO-66-NH_2_@Au_shell_-BNN6 composite nanoparticles (UA-BNN6) entered tumor cells and were irradiated by NIR, the porous gold nanoshells on the surface of composite nanoparticles induced an increase in temperature through the photothermal conversion process and promoted the decomposition of BNN6 molecules, releasing high concentration of NO, thus efficiently killing HeLa cells through the synergistic effect of NO-photothermal therapy. This effective, precise and safe treatment strategy controlled by NIR laser irradiation represents a promising alternative in the field of cancer treatment.

## 1. Introduction

Since the discovery of NO as an important biological signal molecule, it has attracted considerable attention from researchers [1,2]. As a gas transmitter, NO has been found to play an important role in regulating various cellular events, such as vascular dilation, platelet aggregation and adhesion, inflammatory response, immune response, and neurotransmission. Due to its specific properties, NO has great clinical value in cardiovascular diseases, wound healing, and antibacterial and tumor therapy [3,4,5,6,7,8,9]. In particular, NO gas therapy provides an efficient and green therapy method for the treatment of malignant tumors that cause great harm to human health. Due to the constant efforts of scientific researchers, we have a more profound understanding of the mechanism of NO-killing tumor cells. It was found that a high concentration of NO (1 > uM) could induce the apoptosis of tumor cells [10]. The mechanisms behind this apoptotic effect are the ability of NO to induce an oxidative and nitrifying stress, damage mitochondria and DNA, inhibit DNA synthesis and repair, deaminate DNA, and inhibit cell respiration. Although NO has a strong anti-tumor effect, its short life cycle and sensitivity to biological substances limit its clinical applications. Therefore, a major challenge faced by researchers is to develop an effective NO donor and NO-encapsulated cargo carrier to control the release of NO in space, time, and measurement [11,12].

In view of the above problems, various exogenous NO donors have been reported, such as diazeniumdiolates [13], s-nitrosothiols [14], metal-nitrosyl complexes [15], and nitrobenzene derivatives [16]. However, they often have the disadvantage of spontaneous uncontrolled release, which seriously hinders their clinical application. Usually these NO donors are attached or immobilized on small drug molecules, organic polymer materials, or inorganic materials [17,18,19,20]. At the same time, in order to efficiently release NO in tumor tissues, such NO delivery platforms can generally release NO in response to specific stimuli, such as light, heat, or pH [21,22,23,24]. Among several methods, photoexcitation stands out for its ease of operation and safety. Currently, the main light source applied is ultraviolet and visible light. However, this kind of light source has the disadvantages of shallow tissue penetration and significant side effects on surrounding tissues, whereas near-infrared light represents the best choice due to its deep tissue penetration and low phototoxicity [25,26]. In recent years, the NO release platform based on near infrared light excitation has been developed, and relatively ideal experimental results have been obtained.

MOF-NH_2_@Au_shell_ composite nanomaterials not only have the advantages of good biocompatibility, easy degradation, and easy chemical modification of MOF materials [27,28] but also possess excellent photothermal properties and highly porous gold nanoshells [29,30]. Compared with SiO_2_@Au_shell_ materials coated with a dense gold nanoshell [31,32], MOF-NH_2_@Au_shell_ composite nanomaterials can be easily functionalized due to their richness of material selection, the controllability of their size and structure, and the porosity of their gold shell. MOF-NH_2_@Au_shell_ also has great advantages in terms of drug delivery and controlled release. Here, we used the UA as a carrier to immobilize the NO donor reagent, BNN6, obtaining a multifunctional UA-BNN6. The surface plasmon resonance properties of the gold nanoshell surface layer of the UA endow the material with excellent photothermal properties, whereas its rich hierarchical pore structure provides space for the loading of BNN6. The BNN6 molecule has low cytotoxicity and is easily degraded to release NO at high temperatures. It is considered to be an ideal peripheral NO donor. By combining the photothermal conversion performance of UA with the decomposition and release of NO by BNN6 at high temperature, the synergistic effect of NO-photothermal therapy on tumor cells initiated by infrared NIR response was achieved [33,34]. After the composite nanomaterial entered cancer cells, the NO donor molecule was successfully transported inside the cells. Under NIR irradiation, the nanomaterial generated a high temperature and BNN6 was degraded to a released high concentration of NO. In addition, the temperature increase induced by the composite nanomaterials also caused damage to cancer cells (Figure 1). Compared with photothermal therapy alone, the synergistic effect of NO-photothermal therapy has the advantages of using low dosage, being simple to operate, and having few side effects. This green and efficient cancer treatment strategy has great clinical application value.

## 2. Materials and Methods

### 2.1. Reagents and Instruments

Thiazolyl blue (MTT) and N,N′-Bis(1-methylpropyl)-1,4-phenylenediamine (BPA) (95%) were purchased from Sigma-Aldrich (St. Louis, MO, USA). Zircomiun tetrachloride (98%), 2-aminoterephthalic acid (98%), N,N-Dimethylformamide(DMF), potassium carbonate, hydrochloric acid (HCl), chloroauric acid (HAuCl_4_), polyvinyl pyrrolidone (PVP, Kr = 40,000), sodium nitrite (≥99%), and other reagents were of analytical grade and were purchased from Sinoreagent (Shanghai, China). The Hoechst 33342, nitric oxide assay kit, and NO fluorescent probe DAF-FM DA were purchased from Beyotime Biotechnology Co. Ltd. (Nanjing, China). Phosphate buffer saline (PBS) and all other solutions were prepared throughout using ultrapure water (18.2 MΩ cm^−1^, Millipore, MA, USA).

The scanning electron microscope images, energy dispersive spectrometer (EDS), and EDS mapping images were acquired with a Zeiss Ultra Plus SEM (Zeiss, Oberkochen, Germany). The fluorescence and UV−vis spectra were recorded using a Shimadzu RF-5301PC fluoremeter (Kyoto, Japan) and a Biomate 3S spectrophotometer (Thermo Fisher, MA, USA), respectively. The FT-IR spectrum was performed on a Nicolet iS5 IR spectrometer (Thermo Fisher, Waltham, MA, USA) in the range of 400–4000 cm^−1^. The MTT assays were carried out using a MK3 microplate reader (Thermo Fisher, Waltham, MA, USA). The confocal microscopic images were acquired with a Nikon Ti-E confocal microscopy (Nikon, Tokyo, Japan). The NIR laser (λ = 808 nm, maximum power of 1 W/cm^2^) was purchased from Lei Rui Company, Changchun, China. The infrared mappings were imaged by an FLIR C3 infrared camera (FLIR Systems Inc, Wilsonville, OR, USA).

### 2.2. Synthesis of UiO-66-NH_2_ Nanoparticles

UiO-66-NH_2_ nanoparticles were synthesized based on previous reports [29]. Briefly, 40 mg ZrCl_4_ and 31 mg of 2-aminoterephthalic acid were dispersed in 5 mL DMF and then added to 28 μL deionized water. The final mixture was hydrothermally treated at 120 °C for 24 h. The obtained product was washed with DMF and ethanol, respectively. The final product was dispersed in 10 mL of deionized water.

### 2.3. Synthesis of UA

The preparation of growth solution was as follows: 50 mg K_2_CO_3_ was dispersed in 100 mL water, then 1.5 mL of 50 mM HAuCl_4_ aqueous solution was added and stirred at 900 rpm for 10 min. Finally, the mixture was placed in a dark environment for 12 h.

The synthesis of UA was as follows: 5 mL of the prepared UiO-66-NH_2_ mixture was added to 50 mL of the growth solution. Then, 5 mL of 1 mg/mL PVP (MW = 40,000) aqueous solution and 1 mL of formaldehyde were added and stirred at room temperature for 10 min. The reaction solution changed from colorless to dark blue. The final product was washed with dionized water, centrifuged 3 times, and then dispersed in 1 mL of deionized water.

### 2.4. Synthesis of BNN6

Next, 2.34 mL (10 mmol) of N, N′-Bis(1-methylpropyl)-1,4-phenylenediamine (BPA) was diluted in 18 mL of ethanol, and 20 mL (6 M) of degassed NaNO_2_ aqueous solution was added dropwise to the above solution with a separating funnel and stirred for 30 min under nitrogen atmosphere. Subsequently, 20 mL (6 M) of HCl was added dropwise through a separating funnel. As the reaction progressed, the color of the reaction solution gradually changed from red to orange, and off-white precipitate appeared. After stirring for 4 h, the precipitate was collected by centrifugation and then centrifuged and washed with 50% ethanol several times. The final product was freeze-dried and stored at −20 °C, away from light.

### 2.5. Adsorption of BNN6 on UA

Next, 10 mg UA and 10 mg BNN6 were dispersed in 10 mL of DMSO. After stirring at room temperature for 24 h, the obtained solution was washed with DMSO and centrifuged several times to remove free BNN6. The final product was then washed with dionized water and freeze-dried, and the adsorption capacity of BNN6 to the nanoparticles was measured using an ultraviolet-visible spectrometer.

### 2.6. Cytotoxicity test of UA

The cytotoxicity test of UA was evaluated by MTT. Briefly, 4000 Hela cells per well were inoculated into 96-well plates and cultured in DMEM medium. After incubating for 12 h at 37 °C, the medium was removed, and 100 μL of fresh medium containing different concentrations of samples was added to each well, respectively. After another 24 h of incubation, medium was discarded, and 100 μL serum-free medium containing MTT (0.5 mg/mL) was added. After a further incubation of 4 h at 37 °C, medium was removed, and 150 μL DMSO was added to each well. After shaking on a shaker for 10 min, the absorption was detected at 490 nm using a microplate reader.

### 2.7. Cellular Uptake Study

The synthesis UA–FIT (15 mg of UA and 5 mg of FITC) was dissolved in 5 mL DMSO. After the mixture was stirred at room temperature for 24 h, the obtained product was centrifuged and washed several times using DMSO and deionized water and then dispersed in deionized water.

Cell uptake of composite nanoparticles: to monitor the uptake of nanomaterials by cells, we used UA–FITC to observe the cells under a laser confocal microscope. First, HeLa cells were inoculated in a confocal dish and cultured with DMEM at 37 °C for 12 h. Subsequently, 100 μL of UA–FITC in DMEM solution (100 μg/mL) was added, and cells were incubated for another 6 h. The medium was then removed, and cells were washed with PBS (pH 7.4) three times to remove the non-uptaken nanoparticles. Finally, the cell nucleus was labeled with Hoechst 33342 dye, and cells were visualized using a confocal laser scanning microscope (Nikon, Tokyo, Japan) under a 60-fold oil immersion lens.

### 2.8. Photothermal Properties of UA

The photothermal properties of nanoparticles were evaluated by infrared photothermal imager (FLIR Systems Inc, Wilsonville, OR, USA), and 1 mL of a certain amount of UA dissolved in PBS was placed in a centrifuge tube. The solution was then irradiated with a near-infrared laser (Lei Rui, Changchun, China) (1 W. cm^−2^) with a wavelength of 808 nm, and the temperature was recorded every 30 s with an infrared thermal imaging camera (FLIR Systems Inc, Wilsonville, OR, USA).

### 2.9. Photothermal Stability of UA

Using the same above-mentioned procedure, l mL of PBS containing 100 μg of UA was placed in a centrifuge tube and irradiated with NIR laser (Lei Rui, Changchun, China) (808 nm, 1 W/cm^2^) for 5 min, followed by natural cooling for 15 min. Temperature changes were recorded by infrared thermal imager every 30 s. The above procedure was repeated 5 times.

### 2.10. NO Release Performance of UA-BNN6

Based on the phenomenon that BNN6 molecules decompose and release NO at high temperatures, the photothermal conversion performance of the gold nanoshells present on the surface of the composite nanoparticles under NIR radiation provides conditions for the release of NO, thus ensuring this NO release process is controlled by NIR switch on/off. The experimental procedures for the evaluation of the NO-releasing performance of our nanomaterials are as follows: the PBS solution and a certain concentration of different samples were prepared and exposed to NIR (808 nm) with a certain energy density for a certain period of time. Subsequently, the NO concentration and fluorescence intensity of the as-prepared solutions in each group were measured using a nitric oxide detection kit (Beyotime Biotechnology, Nanjing, China) according to the manufacturer’s instructions. DAF-FM DA was used as a NO fluorescence probe.

### 2.11. Intracellular NO Fluorescence Detection

The presence of NO in HeLa cells was observed using DAF-FM DA as a NO fluorescent probe. HeLa cells were inoculated into 6-well plates and incubated with DMEM medium at 37 °C for 12 h. The medium was then removed and DMEM medium containing UA-BNN6 or UA (100 μg/mL) was added. After further incubation for 6 h, an appropriate volume of diluted DAF-FM DA was added and incubated for 20min for staining. Finally, the cells were washed with PBS (pH 7.4) three times to remove the non-uptaken DAF-FM DA, and irradiation was continued with NIR for a certain time (0–30 min). Cell fluorescence was observed by confocal laser scanning microscopy (LSCM) under excitation wavelength of 488 nm.

### 2.12. Determination of Intracellular NO Concentration

The concentration of NO in HeLa cells was determined using a nitric oxide detection kit purchased from Biyuntian. The specific steps were as follows: first, HeLa cells with a concentration of 1 × 10^5^ cells/well were seeded in 6-well plates and incubated at 37 °C for 12 h with DMEM medium. The medium was then removed, and DMEM medium containing UA-BNN6 or UA (100 μg/mL) was added. After further incubation for 6 h, cells were irradiated with NIR for a certain period of time (0–30 min) and then collected. Finally, nitric oxide assay kit was used to determine the concentration of NO in cells as per the manufacturer’s instructions.

### 2.13. In Vitro NO-Photothermal Therapy Synergistic Effect against Cancer Cells

The NO-photothermal synergistic treatment efficiency of composite nanoparticles on tumor cells was evaluated using MTT assay. HeLa cells were seeded in a 96-well plate (100 μL/well) at a density of 1 × 10^4^ cells/well. After incubating at 37 °C for 24 h, the medium was removed. Then, 100 μL of fresh medium containing different samples was added to each well. After another 4 h incubation, some of the wells were irradiated with 808 nm NIR laser (1 W·cm^−2^) for 20 min. After another 24 h of incubation, the medium was removed, 100 μL of serum-free medium containing MTT (0.5 mg/mL) was added, and plates were placed it in a 37 °C incubator for 2 h. The liquid was then discarded, and 150 μL of DMSO was added to each well and placed on a shaker for 10 min, under dark conditions. Finally, the absorbance was measured at 490 nm with a microplate reader.

### 2.14. Animal Model

BALB/c female nude mice (5–6 weeks old) were purchased from Nanjing Yunqiao Purui Biotechnology Co., Ltd. (Nanjing, china) and were given *ad libitum* access to food and water. All animal experiments were conducted in accordance with the guidelines of the Southeast University Animal Research and Ethics Committee and approved by the National Institute of Biological Sciences and the Southeast University Animal Care Research Advisory Committee. HeLa cells were inoculated subcutaneously into the right side of each nude mouse to obtain HeLa xenograft-bearing mice.

### 2.15. In Vivo Antitumor Efficacy Study

The in vivo antitumor efficacy of UA-BNN6 was evaluated using HeLa tumor-bearing mice (23–28 g) as animal model. When tumors reached a size of 100 mm^3^, nude mice were randomly divided into five groups: PBS (control group), UA, UA-BNN6, UA +NIR group and UA-BNN6+NIR group. PBS (150 uL), UA and UA-BNN6 (2 mg ml^−1^, 150 uL) groups received an intratumoral injection. NIR laser irradiation (808 nm) was set at 1.0 W cm^−2^ for 15 min. A second treatment was repeated 6 days after the first one. The weight and tumor size of nude mice were recorded every two days. The volume was calculated using the following formula: V = d^2^ × D/2 (d is the shortest diameter of the tumor and D is the longest diameter of the tumor). After 14 days of treatment, nude mice were euthanized and tumors were removed and weighed. Finally, the main organs of nude mice were excised and stained with hematoxylin and eosin (H&E) for histological analysis.

## 3. Results and Discussion

In this study, UA were synthesized by mixing amino group-rich UiO-66-NH_2_ nanoparticles with HAuCl_4_ and K_2_CO_3_.The deposition of gold seeds on the surface of the composite nanoparticles takes place through electrostatic adsorption with the amino groups part, whereas the gold nanoshells with porous structure is generated under the reduction action of formaldehyde [29]. UiO-66-NH_2_ nanoparticles were obtained by the coordination of ZrCl_4_ with the organic ligand amino terephthalic acid. As shown in Figure 1a, the TEM image revealed the relatively uniform size (particle size of about 60 nm) of the synthesized nanoparticles, with a typical octahedral crystal morphology of UiO-66 nanocrystals. In this study, we used UiO-66-NH_2_ nanocrystals as a template to further synthesize UA. TEM images (Figure 1b,c) clearly show that the synthesized nanoparticles were homogenous, with a particle size of about 80 nm, indicating that the thickness of the gold nanoshells of the as-prepared nanoparticles is about 10 nm. The obtained nanoparticles still retained the crystalline morphology of UiO-66-NH_2_. In addition, it can be clearly observed that the composite nanomaterials have a hierarchical porous structure, composed of the pore structure in the center of UiO-66-NH_2_ and the large pore channel structure of the gold nanoshells on the surface. Furthermore, the EDS spectra (Figure 1d) of the nanoparticles showed obvious peaks of the Au element. EDS mapping image (Figure 1e) of the same sample also confirms that the material contained a high amount of the Au element, as high as 73.46%. The above results fully demonstrate that UA with a hierarchical pore structure have been successfully synthesized.

Benefiting from the advantages of the abundant hierarchical pore structure and the high specific surface area, UA become an ideal carrier with efficient loading capacity. In this study, BNN6 was efficiently loaded into the pores of UA through π-π force and hydrogen bonding. The synthesis of the NO donor BNN6 was obtained by reacting BPA and NaNO_2_ in an acidic environment based on existing reports, and its synthesis path is shown in Appendix A. Subsequently, UA were placed in a DMSO solution of BNN6 and stirred overnight to obtain UA-BNN6. The infrared spectroscopy of each sample (Figure 2a) shows that UA-BNN6 had an obvious absorption peak at 1377 cm^−1^, which is attributed to the stretching vibration peak of N-N = O, whereas no obvious absorption peak was seen in the infrared spectrum of UA. These results indicate that BNN6 was successfully adsorbed on UA-BNN6. In addition, it was calculated from the UV-vis spectra (Appendix A) of the supernatant that when the concentration of BNN6 was 1 mg/mL, the loading amount of BNN6 into the nanoparticles was 2.8 mg/5 mg (Figure 2b). Finally, the SEM image (Figure 2c) of UA-BNN6 showed that the crystal morphology of the material did not change after adsorption of BNN6 molecules.

It is well known that gold nanomaterials have excellent photothermal conversion performance due to their surface plasmon resonance properties. In addition, their excellent photothermal stability and good biocompatibility render them a great thermal sensitizer, which has been widely used in the domain of photothermal therapy of cancer treatment. Similarly, UA have excellent photothermal properties due to the presence of gold nanoshells on their surface. According to the UV-vis spectrum of UA (Figure 3b), it can be seen that it has a strong oscillation absorption peak around 808nm, which indicates that our synthesized nanomaterials would have an excellent photothermal conversion performance under laser irradiation in the near-infrared region. Subsequently, we used an infrared thermal imager to evaluate the photothermal conversion performance of UA under 808 nm NIR laser irradiation. As shown in Figure 3c, when a PBS solution of UA (100 μg/mL) is irradiated by 808 nm NIR laser (1.0 W·cm^−2^), the temperature increases rapidly with the increase of irradiation time. The temperature increased up to 64 °C after 300 s of irradiation. In contrast, the temperature of the free PBS solution in the control group remained at about 25 °C under the same conditions of laser irradiation. In addition, PBS solutions containing different concentrations of UA were irradiated by NIR laser under the same conditions (Figure 3a), and temperature changes were observed by infrared thermal imager. It can be seen that the change in temperature significantly depend on the concentration and irradiation time. The photothermal stability of the material was evaluated by controlling the heating/cooling cycle of the material by switching the laser on and off. The results show (Figure 3d) that the photothermal conversion performance of our synthesized nanomaterials hardly decreases even after the sample has been subjected to the repeated heating and cooling process five times. The above results fully demonstrate that UA have excellent photothermal properties and are promising candidates as photothermal agents.

Studies have reported that BNN6 can stimulate decomposition and release NO under ultraviolet light or a high-temperature environment. In addition, its low cytotoxicity makes it widely used as a NO donor reagent for NO gas therapy. The gold nanoshell layer on the surface of UA-BNN6 has high photothermal conversion performance and generates local high temperature under the irradiation of NIR laser. This provides favorable conditions for the decomposition of BNN6 molecules loaded on the material to release NO. This strategy of releasing NO in response to NIR light provides the possibility for the precise and controllable release of NO. In vitro NO release performance of our nanomaterials was evaluated using a nitric oxide detection kit and NO fluorescence probe DAF-FM DA. Figure 4a shows that when the concentration of composite nanoparticles is 100 μg/mL, the release rate of NO showed an obvious dependence on the NIR laser energy density, with most of NO is released after a 15 min NIR laser irradiation time. As expected, a release of NO can be hardly detected in the control group under the same conditions, which indicates that the NIR laser irradiation does not cause the decomposition of BNN6 molecules. This is consistent with the results of the UV-vis spectra (Appendix A), which showed an obvious absorption oscillation peak in the 365 nm region and a very weak absorption oscillation peak in the near-infrared region. Furthermore, to monitor the release of NO, UA-BNN6 were heated directly at 50 °C or continuously irradiated with an 808 nm near-infrared laser (1.0 W/cm^2^). The results showed (Figure 4b) that the release of NO reached 4.62 μM when the NIR laser irradiation lasted for 20 min, while only a very small amount of NO was detected when the sample was directly heated. This indicates that photothermal agents can greatly improve the decomposition efficiency of BNN6 and release NO because the temperature on the material surface is much higher than the average temperature of the medium [35]. In addition, Figure 4c shows that the NO release behavior of UA-BNN6 is controlled by switching the NIR laser on or off. Upon turning on the NIR illumination, NO release from our nanomaterial is initiated. Conversely, when the NIR light source is turned off, the NO release from the nanomaterial stops immediately. Correspondingly, the release of NO was detected by NO fluorescence probe DAF-FM DA. Appendix A shows that only the fluorescence intensity of the PBS solution of UA-BNN6 was significantly enhanced after NIR laser irradiation, showing obvious concentration Appendix A and time Appendix A dependence. The fluorescence intensity of the sample was basically stable after laser irradiation for 20 min, indicating that BNN6 molecule was basically decomposed completely. In the corresponding photographs (Appendix A), it can be observed that with the increase of NIR laser irradiation time, the color of the mixture gradually changes from white to wine red, which is due to the decomposition of the beige BNN6 molecule into the wine red BHA molecule and NO at high temperatures.

The biocompatibility of UA was evaluated using the MTT assay. The results (Figure 5a) show that even if UA concentration reaches 50 μg/mL, the cell survival rate of HeLa cells is still as high as 87.7%, indicating that our nanomaterial has low cytotoxicity. Subsequently, UA–FITC obtained by replacing BNN6 molecules with FITC molecules was co-incubated with HeLa cells, and the nuclei were labeled with Hoechst 33342 dye. Finally, a laser scanning confocal microscope (LSCM) was used to observe the uptake or our nanomaterial by HeLa cells with a 60× oil-immersion objective. As displayed in Figure 5b, strong green fluorescence appears in the cytoplasm, while only blue fluorescence appears in the nucleus, indicating that the material mainly enters the cytoplasm but not the nucleus after being uptaken by cells. These results fully indicate that nanoparticles have good biocompatibility and cell uptake.

Similarly, we used HeLa cells to investigate the intracellular NO release performance of UA-BNN6. The NO concentration in the cells was measured using a nitric oxide detection kit, and the cells were observed with the NO fluorescent probe DAF-FM DA. The fluorescence images (Appendix A) of each cell group showed that the cells only showed weak fluorescence after co-incubation with the sample UA-BNN6 or UA. Similarly, the cells incubated with the UA and irradiated with an NIR laser also showed weak fluorescence. In contrast, when the cells were incubated with the sample UA-BNN6 and irradiated with a NIR laser for 20 min, the fluorescence image showed strong fluorescence. Correspondingly, the NO concentration in the cells of each experimental group was determined using a nitric oxide detection kit. The results (Figure 6a) showed that after the cells were co-incubated with UA-BNN6 and irradiated with NIR laser, the concentration of NO was 3.5 times that of the control group. The concentration of NO in the other experimental groups was almost unchanged compared with the control group. Furthermore, the intracellular concentration of NO was determined under different NIR laser irradiation times. The results (Figure 6b) showed that the intracellular concentration of NO basically remained stable after continuous NIR laser irradiation for more than 20 min with (808 nm, 1.0 W/cm^2^). This shows that our composite nanoparticles can promptly respond to NIR laser stimulation to release most of the NO. Correspondingly, the cells labeled with the NO fluorescent probe DAF-FM DA (Figure 6c) showed that the fluorescence intensity of the cells was obviously dependent on the duration of NIR laser irradiation, and the fluorescence intensity was basically stable after 20 min. These results further indicate that UA-BNN6 taken up by cells can respond to the stimulation of NIR light and release a large amount of NO in a precise and controllable manner.

Given the fact that BNN6 molecules degrade and release NO at high temperatures, and with the excellent photothermal properties of the gold nanoshells present on the surface of UA-BNN6 upon exposition to NIR laser irradiation, our synthesized nanomaterial can not only decompose and create a high-temperature environment for BNN6 but also lead to a local temperature increase in the cancer cells, finally achieving a synergistic NO-photothermal anti-cancer effect. Herein, using HeLa cells as an in vitro model, an MTT assay was used to evaluate the NO-photothermal synergistic anticancer properties of UA-BNN6. The results (Figure 6d) showed that the tumor cells maintained a high survival rate even when high concentrations of UA or UA-BNN6 were used, indicating the low cytotoxicity of the composite nanoparticles. On the contrary, cells co-incubated with the sample UA or UA-BNN6 and irradiated with a 808 nm NIR laser (1.0 W/cm^2^) showed significant cytotoxicity. At the concentration of 50 μg/mL, each group showed different cytotoxicity results. Among them, only tumor cells incubated with UA or UA-BNN6 and irradiated with NIR laser exhibited significant cytotoxicity, where UA-BNN6 showed the most severe cytotoxicity. The former showed strong cytotoxicity, mainly due to the photothermal effect of gold nanoshells, while the latter benefited from the NO-photothermal synergistic antitumor effect of the material. Using composite nanomaterials that can respond to NIR stimulation and quickly initiate the process of NO-photothermal synergistic killing of cancer cells is an efficient and green anti-tumor method.

In view of the satisfactory synergistic killing effect of UA-BNN6 composite nanoparticles on cancer cells, HeLa tumor-bearing mice were used as animal models to evaluate the antitumor properties of the composite nanoparticles in vivo. Firstly, nude mice were injected with UA-BNN6 or PBS and irradiated with near-infrared laser (808 nm, 1.0 W/cm^2^) for 5 min. To evaluate the photothermal efficiency of the material in nude mice, infrared thermal imaging system was used to observe the changes of surface temperature. As displayed in Figure 7a,b, the temperature at the tumor site of the nude mice injected with UA-BNN6 increased significantly upon laser irradiation, reaching 44.9 °C just after 1 min of irradiation and further rising to 54.3 °C after 5 min. In contrast, nude mice injected with PBS showed only a slight increase in temperature after laser irradiation for 5 min. These results indicate that the UA-BNN6 can be used as a thermosensitive agent for in vivo tumor photothermal therapy. Subsequently, to further verify the therapeutic performance of the composite nanoparticles on tumors, we randomly divided HeLa xenograft-bearing mice (with an initial tumor volume of 100 mm^3^) into five groups: PBS group (control group), UA group, UA-BNN6 group, UA+NIR group, and UA-BNN6+NIR group. Nude mice in each group received corresponding treatment every 6 days. The results showed (Figure 7e) that the tumors of nude mice in the PBS, UA, and UA-BNN6 groups increased significantly within the first 14 days, mainly due to the low tumor-growth inhibition effect of the administrated samples. In contrast, tumors in nude mice of the UA+NIR and UA-BNN6+NIR groups were significantly shrunken and practically disappeared. However, it was evident that the wound healing of nude mice treated with UA+NIR was significantly slower than that of the UA-BNN6+NIR group. The former could inhibit tumor growth mainly through photothermal effect upon NIR irradiation. Indeed, the NIR laser irradiation of nude mice in the UA-BNN6+NIR group caused an increase in temperature in the tumor area and promoted the decomposition of the NO donor, BNN6, thus releasing a large amount of NO. Compared with the single PTT treatment, UA-BNN6+NIR achieved a remarkable NO-photothermal synergistic effect, resulting in a more efficient anti-tumor effect. Indeed, various studies have reported that NO can promote wound healing [36]. The tumor volume growth (Figure 7c) recorded in nude mice during the treatment period also showed that the most effective tumor growth inhibitory response was obtained in the UA+NIR and UA-BNN6+NIR groups. After 14 days of treatment, the tumors of the nude mice were removed and weighed, and the results (Figure 7e) showed that the final tumor weights were consistent with the tumor growth behavior in each group. In addition, no significant decrease in body weight of nude mice during different treatments was observed (Figure 7d), indicating the high biocompatibility of the composite nanoparticles. Finally, major organs (heart, liver, spleen, lung, and kidney) of nude mice were excised and stained with hematoxylin and eosin (H&E) for histological analysis. The results showed (Appendix A) that, compared with the PBS group, no obvious damage was seen in all major organs in each group, further confirming the excellent biocompatibility of the nanomaterial. The above results fully demonstrate that UA-BNN6 are highly efficient, biocompatible, and safe nanomaterials for synergistic NO-photothermal therapy of cancer, hence having outstanding application prospects as anti-tumor agents to realize targeted cancer clearance.

## 4. Conclusions

In summary, in this contribution UA with excellent NIR absorption properties were successfully synthesized by a simple and gentle method. Owing to the rich porous structure of the synthesized composite nanoparticles, UA can efficiently deliver the hydrophobic NO donor BNN6 into cancer cells/tissues. Based on the phenomenon that BNN6 molecules can decompose and release NO at high temperatures, UA-BNN6 that enter cancer cells can generate local high temperatures under NIR irradiation and promote the decomposition of BNN6 molecules to release relatively high concentrations of NO, so as to achieve a NO-photothermal therapy synergistic effect against cancer cells. This NIR-controlled treatment strategy has the advantages of high anti-cancer efficacy, precision, and good biocompatibility, paving a way for a future green and efficient cancer therapy approach for targeted cancer clearance.

## Data Availability

Not applicable.

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
