# Peer review of "Near-Infrared Light-Triggered Nitric Oxide Nanogenerators for NO-Photothermal Synergistic Cancer Therapy"

_nanomaterials, 2022, doi:10.3390/nano12081348_

Round 1

Reviewer 1 Report

The manuscript by Liu et al. deals with the development and further testing of a new nanoplatform for cancer treatment. The paper topic is fully within the scope of the Nanomaterials journal; the results presented are certainly interesting for the wide readership of the journal. The paper is well structured, the material is presented in a consistent way, the conclusions are adequately supported by the results.

For the first time, amino-functionalized MOF (UiO-66-NH2) were used as a template for MOF@Au composite nanoparticles preparation with high loading capacity and excellent photothermal properties. It was clearly justified that the composite obtained can be heated locally upon NIR irradiation generating NO molecules which are deleterious to cancer cells. This paper provides a major advance towards targeted cancer therapy by nanoplatforms. This paper can be published in the Nanomaterials journal with some minor corrections.

  1. The novelty of the work is poorly reflected in the Abstract, Introduction and Results sections. The synergistic cancer therapy approach proposed by the authors is not unique [10.1016/j.jconrel.2022.03.030]. A comparison is needed with the other composites proposed earlier for synergistic cancer therapy. What is the reason for MOF template use?
  2. What do the authors mean by “green” therapy methods in the Introduction section? It is advisable to clarify in more detail why the method proposed by the authors refers to the “green” therapy methods.
  3. In section “2.2. Synthesis of UiO-66-NH2 Nanoparticles”, the sentence "The final mixture was added to a high pressure reaction kettle and reacted at 120 oC for 24 h." should be replaced with "The final mixture was hydrothermally treated at 120 oC for 24 h."
  4. In the Results and discussion section, the authors mentioned that the gold content in the UiO-66-NH2@Aushell composite reaches 73.84 %. Is this weight or atomic percents? Please provide information on Zr:Au ratio in UiO-66-NH2@Aushell composite.
  5. How did the authors confirm that they obtained the UiO-66-NH2 MOF? What is the crystal structure of UiO-66-NH2 (space group, lattice parameters, topology)?
  6. In the Results and discussion section, on page 7, the authors mentioned high specific surface area of UiO-66-NH2. Was the specific surface area measured in the work?
  7. The loading of BNN6 should be presented as BNN6/UiO-66-NH2@Au shell mass ratio, e.g. 3 mg per 5 mg.
  8. Please specify the purity of the precursor reagents.

Author Response

The manuscript by Liu et al. deals with the development and further testing of a new nanoplatform for cancer treatment. The paper topic is fully within the scope of the Nanomaterials journal; the results presented are certainly interesting for the wide readership of the journal. The paper is well structured, the material is presented in a consistent way, the conclusions are adequately supported by the results.

For the first time, amino-functionalized MOF (UiO-66-NH2) were used as a template for MOF@Au composite nanoparticles preparation with high loading capacity and excellent photothermal properties. It was clearly justified that the composite obtained can be heated locally upon NIR irradiation generating NO molecules which are deleterious to cancer cells. This paper provides a major advance towards targeted cancer therapy by nanoplatforms. This paper can be published in the Nanomaterials journal with some minor corrections.

Point 1: The novelty of the work is poorly reflected in the Abstract, Introduction and Results sections. The synergistic cancer therapy approach proposed by the authors is not unique [10.1016/j.jconrel.2022.03.030]. A comparison is needed with the other composites proposed earlier for synergistic cancer therapy. What is the reason for MOF template use?

Response 1: Thanks to your kind comments. Metal-organic frameworks (MOFs) coordination compounds consisting of metal ions/clusters and organic ligands, intrinsic biodegradability, facile and designable synthesis on the nanoscale, inherent pore apertures and highly enriched functionalities and well-defined tunable sizes/shapes. In addition, the surface of NH2-MOFs synthesized by 2-aminoterephthalic acid as organic Ligands has abundant amino groups, which enables gold seeds to be grafted onto its surface by electrostatic adsorption, followed by reduction to yield gold nanoshell. It is an ideal template for the coating of gold nanoshells by a more facile and straightforward one-step method. The composite nanomaterial with both MOF and gold nanomaterial has drawn tremendous attention in biomedicine due to their superior properties, including the pore apertures for drug loading and controllable release(Reference 29).

Point 2: What do the authors mean by “green” therapy methods in the Introduction section? It is advisable to clarify in more detail why the method proposed by the authors refers to the “green” therapy methods.

Response 2: Thanks to your kind comments. In this experiment, the process of photothermal-No gas co-therapy was activated by near-infrared light. Under the irradiation of near-infrared laser at 808 nm, the efficient photothermal conversion performance of gold nanoshells caused local high temperature in the tumor tissue, and at the same time, the high temperature environment caused the decomposition of NO donor BNN6 to release NO, and finally the photothermal-NO gas co-therapy for the tumor. Among them, near infrared light has high tissue penetration and low phototoxicity, which can perform precise photothermal therapy on tumors, killing tumor tissues efficiently without destroying normal tissues. In addition, the emerging NO gas therapy has been recognized as a “green” treatment paradigm with negligible side effects (Reference 4). This precise treatment of tumor tissue with low side effects can be regarded as a "green" cancer treatment.

Point 3: In section “2.2. Synthesis of UiO-66-NH2 Nanoparticles”, the sentence "The final mixture was added to a high pressure reaction kettle and reacted at 120 oC for 24 h." should be replaced with "The final mixture was hydrothermally treated at 120 oC for 24 h."

Response 3: Thanks for your kind suggestion. We have replaced the sentence "The final mixture was added to a high pressure reaction kettle and reacted at 120 oC for 24 h." with "The final mixture was hydrothermally treated at 120 oC for 24 h." Please see in the revised manuscript.

Point 4: In the Results and discussion section, the authors mentioned that the gold content in the UiO-66-NH2@Aushell composite reaches 73.84 %. Is this weight or atomic percents? Please provide information on Zr:Au ratio in UiO-66-NH2@Aushell composite.

Response 4: Thanks for your kind suggestion. The gold content in the UiO-66-NH2@Aushell composite reaches 73.84 % is this weight percents. From the EDS characterization results (Fig. 1d)of the samples, it can be seen that the atomic ratio of Zr:Au is 1.31:16.7.

Point 5: How did the authors confirm that they obtained the UiO-66-NH2 MOF? What is the crystal structure of UiO-66-NH2 (space group, lattice parameters, topology)?

Response 5: Thanks to your kind comments. In this experiment, we adopted a most typical synthesis method of UiO-66-NH2 nanoparticles (JACS 2008, 130, 13850–13851). Briefly, 40 mg ZrCl4 and 31 mg of 2-aminoterephthalic acid were dispersed in 5 mL DMF and then added to 28 μL deionized water. The final mixture was hydrothermally treated at 120 oC for 24 h. UiO-66-NH2 is a zirconium-based metal-organic framework material with Zr6O4(OH)4 structural unit as the core. In its crystal structure, it includes six inner cores which are interconnected by 2-aminoterephthalic acid to form a regular tetrahedron and regular octahedron with triangular windows connected to each other [Zr6O4(OH)4(BDC-NH2)12]. Each Zr atom in the crystal structure of UiO-66-NH2 is an octacoordinate structure, and the eight oxygen atoms coordinated with it can form a tetragonal antiprism configuration. The four oxygen atoms on one face come from the organic ligand 2-aminoterephthalic acid, and the oxygen atoms on the other face come from μ3-O and μ3-OH groups. The pore sizes of the two pore cages of UiO-66-NH2 are 7.5 Å and 12 Å, respectively, and the pore size of the triangular pore windows that make them interconnected is about 6 Å.

Point 6: In the Results and discussion section, on page 7, the authors mentioned high specific surface area of UiO-66-NH2. Was the specific surface area measured in the work?

Response 6: In this experiment, the synthesis of UiO-66-NH2 nanoparticles is based on the most typical synthesis method of UiO-66-NH2 nanoparticles, and many studies have reported that UiO-66-NH2 nanoparticles have a high specific surface area (1187 m2/g), so the specific surface area of the synthesized material was not tested in this paper (JACS 2008, 130, 13850–13851).

Point 7: The loading of BNN6 should be presented as BNN6/UiO-66-NH2@Aushell mass ratio, e.g. 3 mg per 5 mg.

Response 7: Thanks for your kind suggestion. The loading of BNN6 reach 2.3 mg/5 mg. Please read the revised manuscript for relevant revisions.

Point 8: Please specify the purity of the precursor reagents.

Response 8: Thanks for your kind suggestion. The purity of the precursor reagents has added. Please see it in the revised manuscript.

Reviewer 2 Report

The authors synthesized UiO-66-NH2 MOF and attached Au nanoparticles on the surface of the MOF to prepare UiO-66-NH2@Aushell. Then, they introduced BNN6 (NO generating molecules into UiO-66-NH2@Aushell to prepare UiO-66-NH2@Aushell-BNN6. They used the nanocomposites cancer therapy in vitro using HeLa cells. They also performed in vivo cancer therapy.

  • Could you please describe how the authors prove the Au nanoparticles attachment on the MOF surfaces.
  • Could you please describe how the authors prove the BNN6 inclusion inside the MOFs.
  • Could you please describe the cancer cell killing mechanism by NO.
  • For in vivo tests, cancer-targeting ligands are generally conjugated to nanoparticles. Could you please explain what kinds of cancer-targeting ligands were used. If not used, could you explain why not used?

Author Response

The authors synthesized UiO-66-NH2 MOF and attached Au nanoparticles on the surface of the MOF to prepare UiO-66-NH2@Aushell. Then, they introduced BNN6 (NO generating molecules into UiO-66-NH2@Aushell to prepare UiO-66-NH2@Aushell-BNN6. They used the nanocomposites cancer therapy in vitro using HeLa cells. They also performed in vivo cancer therapy.

Point 1:Could you please describe how the authors prove the Au nanoparticles attachment on the MOF surfaces.

Response 1: Thanks to your kind comments. In this paper, the synthesis of UiO-66-NH2@Aushell composite nanoparticles is based on reports in published literature (Reference 27). Gold seeds are grafted onto the surface of nanoparticles through electrostatic adsorption with amino groups on the surface of UiO-66-NH2 nanoparticles, and then deposited to form gold nanoshells under the reduction of formaldehyde. The TEM images (Fig. 1b, c), EDS spectra (Fig. 1d) and EDS mapping image (Fig. 1e) of the UiO-66-NH2@Aushell sample all indicate the formation of gold nanoshells.

Point 2:Could you please describe how the authors prove the BNN6 inclusion inside the MOFs.

Response 2: Thanks to your kind comments. It can be clearly seen from the IR spectra of samples f BNN6, of UiO-66-NH2@Aushell and UiO-66-NH2@Aushell-BNN6 that NH2@Aushell-BNN6 composite nanoparticles had an obvious absorption peak at 1377 cm-1, which is attributed to the stretching vibration peak of N-N = O. Whereas no obvious absorption peak was seen in the infrared spectrum of UiO-66-NH2@Aushell composite nanoparticles. These results indicate that BNN6 was successfully adsorbed on UiO-66-NH2@Aushell composite nanoparticles.

Point 3:Could you please describe the cancer cell killing mechanism by NO.

Response 3: Thanks to your kind comments. The mechanisms by which high concentrations of NO kill tumor cells include: (1) mediating the tumoricidal effect of macrophages; (2) mediating oncolysis of endothelial cells; (3) combining with intracellular superoxide anion to generate nitrogen/ Oxygen free radicals can damage DNA, resulting in cytotoxicity; (4) affecting the energy metabolism of cells, and tumor cells die due to energy metabolism disorders; (5) inducing tumor cell apoptosis by activating the expression of p53; (6) by inhibiting platelet aggregation and inhibiting tumor metastasis(Reference 4).

Point 4: For in vivo tests, cancer-targeting ligands are generally conjugated to nanoparticles. Could you please explain what kinds of cancer-targeting ligands were used. If not used, could you explain why not used?

Response 4: First of all, I am very sorry for the misstatement in my article. In animal experiments, all samples were injected intratumoral rather than through the tail vein, so the effective treatment of tumors could be achieved even without the modification of tumor targeting ligands. The related misstatement has been corrected in the revised manuscript.

Reviewer 3 Report

The authors need to improve the introduction, abstract and conclusion

Highlight the numerical results in the abstract and conclusion, with appropriate statistical analysis

Upload the raw data for supporting information

Update the literature references to 2020-2022

Author Response

Point 1: The authors need to improve the introduction, abstract and conclusion

Response 1: Thanks for your kind suggestion. Parts of the background, abstract, and conclusions of the manuscript have been revised. Please see it in the revised manuscript.

Point 2: Highlight the numerical results in the abstract and conclusion, with appropriate statistical analysis

Response 2: Thanks for your kind suggestion. Corresponding data results were highlighted in the abstract and conclusion. Please see it in the revised manuscript.

Point 3: Upload the raw data for supporting information

Response 3: First of all, I'm sorry that I don't want my experimental data to be published online, but if a reader asks me to provide data to him or her, I will do so without reservation.

Point 4: Update the literature references to 2020-2022

Response 4: Thanks for your kind suggestion. Some references have been changed to those published in 2020-2022. Please see it in the revised manuscript.

Reviewer 4 Report

This is interesting work describing new nanoparticles which can be used in cancer treatment. The presentation of the results must be improved. In all figures, I suggest separating the pictures in all figures. Moreover, the names of the nanoparticles are too complicated. Could Authors somehow simplify them?

Generally, the paper must be corrected by a native speaker.

Fig 1. The legend must be improved. What are pictures b and c?

Fig 2 pictures are a too small resolution to low, descriptions of  the axes- the fonts are too small

Fig 3 b, c, d  Descriptions of the axes- the fonts are too small. What are the parameters of irradiation in picture 3c?

Fig 4 Again resolution of the fonts describing axes is too small; the same applies to the legends. What about standard deviations there?

Fig 5 there are nice pictures from the microscope; could you make them larger?

Fig 6 This picture is an excellent example of how simplifying nanoparticles' names could increase readability.  Again resolution of text describing the charts is too tiny, microscopic images may be larger, the space bars are lacking. Picture c - descriptions of the pictures in lines is lacking. The power of irradiation should be mentioned

Fig 7 This picture is also a good example of simplifying the names of nanoparticles that could increase readability.  Again, the text describing the charts is too tiny; the picture d - should be “body”, not “boyd”. Picture A is it possible to adjust the scale and start from 35oC? The tumours could be better visible.

The number of the approval from the ethics committee has not been provided.

How was the power of irradiation measured?

How many cells were seeded for the experiment described in paragraph 2.10?

How many cells were grafted to nude mice?

Which type of lamp has been used for irradiation?

Could the Authors provide is the supplementary file to give a picture of the laser system used to irradiate in vitro samples (including 96 well plates) and animals?

Author Response

Point 1: This is interesting work describing new nanoparticles which can be used in cancer treatment. The presentation of the results must be improved. In all figures, I suggest separating the pictures in all figures. Moreover, the names of the nanoparticles are too complicated. Could Authors somehow simplify them?

Response 1: Thanks for your kind suggestion.UiO-66-NH2@Aushell abbreviated as UA, UiO-66-NH2@Aushell-BNN6 abbreviated as UA-BNN6. Please see it in the revised manuscript.

Point 2: Generally, the paper must be corrected by a native speaker.

Response 2: Thanks for your kind suggestion. The revised manuscript was corrected by a native speaker.

Point 3: Fig 1. The legend must be improved. What are pictures b and c?

Response 3: Thanks for your kind suggestion. The figure 1b and c is the TEM of UiO-66-NH2@Aushell. We have added the legend in the revised manuscript.

Point 4: Fig 2 pictures are a too small resolution to low, descriptions of the axes- the fonts are too small

Response 4: Thanks for your kind suggestion. We have modified Figure 2 as you suggested. Please see it in the revised manuscript.

Point 5: Fig 3 b, c, d Descriptions of the axes- the fonts are too small. What are the parameters of irradiation in picture 3c?

Response 5: Thanks for your kind suggestion. We have modified Figure 3 as you suggested. In Fig. 3c, PBS solution or PBS solution of UiO-66-NH2@Aushell-BNN6 composite nanoparticles (100 μg/mL) is irradiated by 808 nm NIR laser (1 W·cm-2). Please see in the revised manuscript.

Point 6: Fig 4 Again resolution of the fonts describing axes is too small; the same applies to the legends. What about standard deviations there?

Response 6: Thanks for your kind suggestion. We have modified Figure 4 as you suggested. Please see it in the revised manuscript.

Point 7: Fig 5 there are nice pictures from the microscope; could you make them larger?

Response 7: Thanks for your kind suggestion. We have modified Figure 5 as you suggested. Please see it in the revised manuscript.

Point 8: Fig 6 This picture is an excellent example of how simplifying nanoparticles' names could increase readability. Again resolution of text describing the charts is too tiny, microscopic images may be larger, the space bars are lacking. Picture c - descriptions of the pictures in lines is lacking. The power of irradiation should be mentioned.

Response 8: Thanks for your kind suggestion. We have modified Figure 6 as you suggested. Please see it in the revised manuscript.

Point 9: Fig 7 This picture is also a good example of simplifying the names of nanoparticles that could increase readability. Again, the text describing the charts is too tiny; the picture d - should be “body”, not “boyd”. Picture A is it possible to adjust the scale and start from 35 oC? The tumours could be better visible.

Response 9: Thanks for your kind suggestion. We have modified Fig.7 as you suggested. If the adjustment scale starts from 35 oC, it may cause the image of the heating part to be inconspicuous.

Point 10: The number of the approval from the ethics committee has not been provided.

Response 10: Thanks for your kind suggestion. The animal experiment was conducted in accordance with guidelines of the Animal Research and Ethics Board of Southeast University and approved by the National Institute of Biological Science and the Animal Care Research Advisory Committee of Southeast University.

Point 11: How was the power of irradiation measured?

Response 11: Thanks to your kind comments. The power of the near-infrared light required for the experiment can be flexibly controlled on the laser transmitter device.

Point 12: How many cells were seeded for the experiment described in paragraph 2.10?

Response 12: Thanks to your kind comments. The experimental part of Paragraph 2.10 is to study the NO release performance of the sample in PBS, and does not involve the cell experiment.

Point 13: How many cells were grafted to nude mice?

Response 13: Thanks to your kind comments. HeLa cells were inoculated subcutaneously into the right side of each nude mouse to obtain HeLa xenograft-bearing mice. Each nude mouse was injected with 1×107 HeLa tumor cells.

Point 14: Which type of lamp has been used for irradiation?

Response 14: In this experiment, we used the 808 nm infrared semiconductor laser (purchased from Lei Rui Company, Changchun, China.) for irradiation.

Point 15: Could the Authors provide is the supplementary file to give a picture of the laser system used to irradiate in vitro samples (including 96 well plates) and animals?

Response 15: It is a pity that no relevant photos were taken during the relevant experiments, but the details of the experimental operation have been explained in detail in the experimental part of the article.

Round 2

Reviewer 3 Report

Accept